# Retrospective Analysis of Water Management in Amsterdam, The Netherlands

**Sannah Peters [1,2], Maarten Ouboter [1], Kees van der Lugt [3], Stef Koop [2,4] and Kees van Leeuwen [2,4,*]**

1 Waternet (Public Water Utility of Amsterdam and Regional Water Authority Amstel, Gooi and Vecht), P.O. Box 94370, 1090 GJ Amsterdam, The Netherlands; Sannah.Peters@waternet.nl (S.P.); Maarten.Ouboter@waternet.nl (M.O.)
2 Copernicus Institute of Sustainable Development, Utrecht University, Princetonlaan 8a, 3508 TC Utrecht, The Netherlands; stef.koop@kwrwater.nl
3 World Waternet, P.O. Box 94370, 1090 GJ Amsterdam, The Netherlands; Kees.van.der.Lugt@waternet.nl
4 KWR Water Research Institute, P.O. Box 1072, 3430 BB Nieuwegein, The Netherlands
* Correspondence: kees.van.leeuwen@kwrwater.nl

**Abstract:** The capital of The Netherlands, Amsterdam, is home to more than 800,000 people. Developments in water safety, water quality, and robust water infrastructure transitioned Amsterdam into an attractive, economically healthy, and safe city that scores highly in the field of water management. However, investments need to be continued to meet future challenges. Many other cities in the world have just started their transition to become water-wise. For those cities, it is important to assess current water management and governance practices, in order to set their priorities and to gain knowledge from the experiences of more advanced cities such as Amsterdam. We investigate how Amsterdam's water management and governance developed historically and how these lessons can be used to further improve water management in Amsterdam and other cities. This retrospective analysis starts at 1672 and applies the City Blueprint Approach as a baseline water management assessment. It shows that developments in water infrastructure and water management have often been reactive in response to various crises. International knowledge exchange, implementation of integrated water resources management, and long-term planning improved the city considerably. We conclude that experiences from the past can be used to meet present and future challenges in many cities across the globe.

**Keywords:** integrated water resources management; City Blueprint; retrospective analysis; water governance; city-to-city learning

## 1. Introduction

In all cities around the world, demographic, technological, economic, and climate trends have shaped the living environment that sustains us [1]. Water systems are increasingly influenced by human factors that may lead to changes in water availability and quality [2,3]. These pressures bring about challenges in the dynamics of cities such as housing, drinking water, solid waste, and wastewater [3]. Fresh water is crucial for sustainable development and is implemented in one of the United Nations (UN) Sustainable Development Goals, i.e., SDG 6: ensure access to water and sanitation for all. While access to water services is usually adequate in cities of the OECD (Organisation for Economic Co-operation and Development) member states, water networks are aging and require refurbishments, in some cases urgently and extensively. Good governance is fundamental in addressing health and environmental challenges that result from poor investments in water infrastructure [4,5]. Infrastructure that is built and conceived years ago may be insufficiently adapted to the current and future circumstances. For example, urban drainage is often not adapted to downpours, which can lead to combined sewer overflows that affect the quality and biodiversity of surface water [5]. Adapting existing infrastructure to meet current and

future conditions is challenging owing to the high costs and the implementation of new technologies in complex systems [4,5]. Therefore, it is important to thoroughly understand a city's water system, water management performance, and water governance capacities in order to select the best practices to address these challenges and seize opportunities that integrated water management can provide.

Scientific knowledge is often developed to address narrowly defined problems and does not provide actionable strategic insights that can help decision-makers achieve their goals and objectives to address challenges of water, waste, and climate change in their city [6]. The gap between water science, policy, and implementation has been widely acknowledged [6,7].

Various historical water management analyses are available, e.g., [8,9]. However, a specific urban water management performance assessment of almost four centuries may provide new insights. The goal of this paper is to analyse how the city of Amsterdam has developed its water management and water governance over a period of about 350 years. We perform a series of retrospective analyses in a systematic manner to show (1) how water management and governance developed historically in the city of Amsterdam, (2) how the resulting insights can benefit the continued improvement of Amsterdam's water management, and (3) how these lessons can help other cities across the world to meet their current and future challenges. Through these insights, this study aims to contribute to urban water management transition literature [3,6,7].

## 2. Amsterdam: State-of-the-Art

The development of Amsterdam, the capital city of the Netherlands, has been strongly intertwined with water for more than 700 years (Figure 1). The city is home to over 800,000 people [10]. Amsterdam was founded at a strategic location on the edge of the river Amstel and close to the North Sea [11]. Even the name of the city reflects its history with water as it refers to the adjacent river Amstel, which terminates in the historic canals of Amsterdam [10].

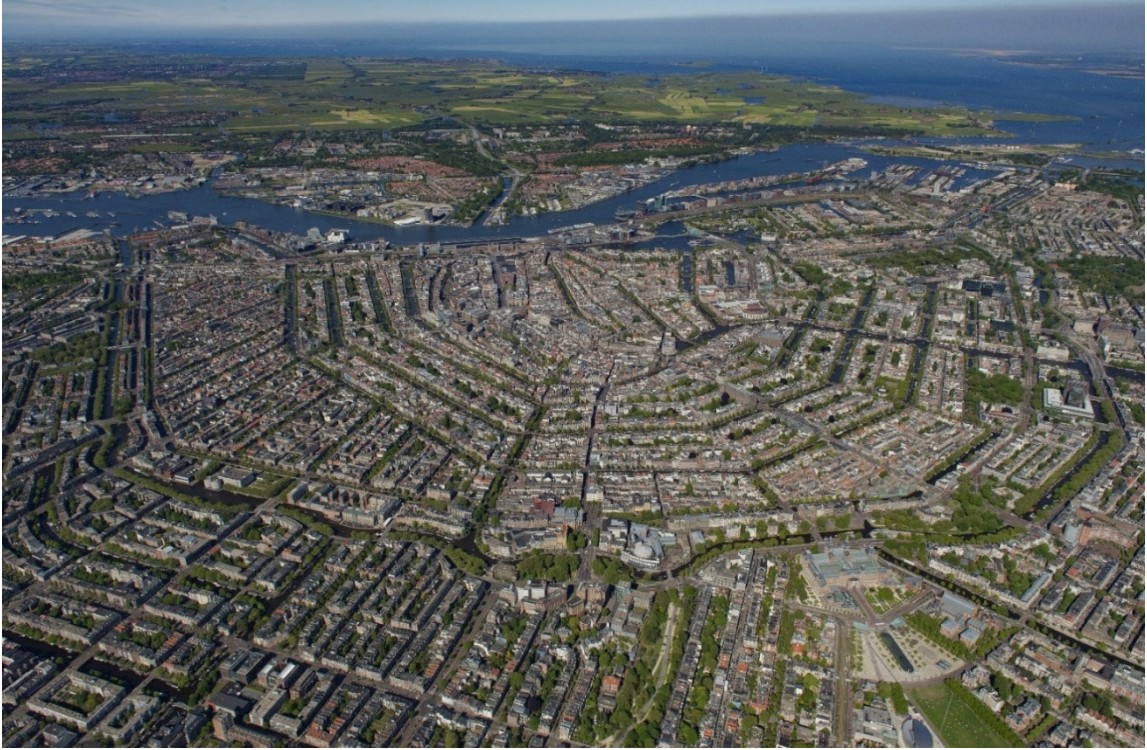

**Figure 1.** Aerial view of the city of Amsterdam showing the urban structure of the 17th century (Source: Office for Monuments and Archaeology, City of Amsterdam, photo by Cor Harteloh; with permission).

Investments in flood risk management, water quality, and robust water infrastructure turned Amsterdam into an attractive, economically healthy, and safe city [11,12]. Amsterdam holds a leading international position in the field of integrated water resources management (IWRM). Accordingly, Amsterdam's Blue City Index, which represents a comprehensive set of 24 indicators that measure integrated water management, is among the highest in the world [12,13]. Owing to pressures related to water (storm surges, intensive precipitation, and drought), a water system has been created that offers a certain degree of protection and security. To keep the water system robust, these efforts, however, need to continue in the dynamic context of demographic, socio-economic, administrative, landscape, and climate developments.

In previous research [10], the City Blueprint Approach was used to examine the sustainability of IWRM of Amsterdam. Since then, the method has been reviewed and updated [12,13]. More recently, a second revision was made to include, among others, the World Bank governance indicators in the Trends and Pressures Framework [14–16]. To date, 125 cities have been assessed. At present, Amsterdam is the best-performing city among the evaluated cities, which comes to expression in the following: (1) the long-term vision and multi-level water governance approach; (2) integration of water, energy, and material flows; (3) the entanglement between urban quality of life and water management; and (4) the open communication to and feed-back from its citizens [12]. The results for Amsterdam are shown in Figures 2 and 3. Both water management in the municipality of Amsterdam and tasks of the Regional Public Water Authority Amstel, Gooi, and Vecht are assigned to a single organisation named Waternet. This water utility is responsible for the public management of integrated water services [11]. It is the only water utility in The Netherlands that covers the whole water cycle [16]. Among other things, the utility is responsible for the supply of drinking water, the maintenance of the sewage system, communication on groundwater levels, surface water management, water quantity and quality, and wastewater treatment [16]. The city's unique water cycle approach has proved to be beneficial [10,16–18].

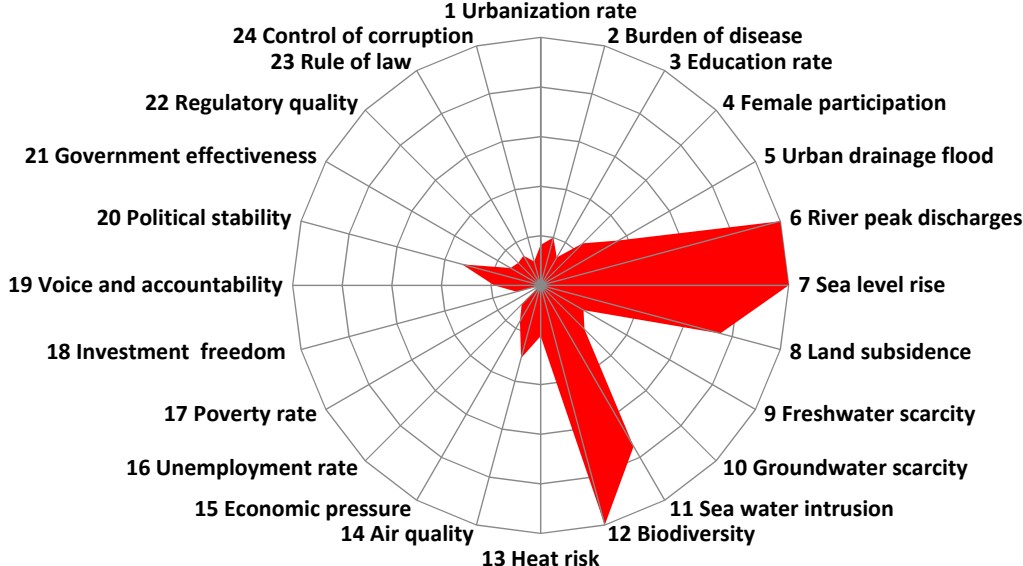

**Figure 2.** Trends and Pressures Framework of Amsterdam (2020) assessed according to [14].

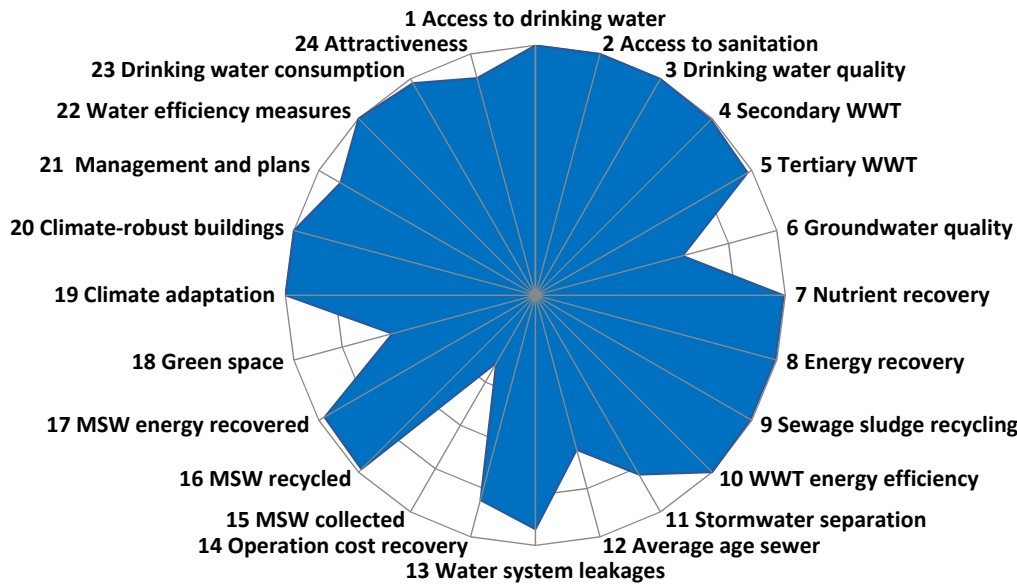

**Figure 3.** City Blueprint of Amsterdam (2020), assessed according to [15]. WWT, wastewater treatment.

While the city ranks high in water management and governance, there is room for further improvement. New investments are necessary to meet future challenges [18]: "Water is indispensable. For us, as inhabitants, and for our companies. Water can also be a threat from which we must protect ourselves. At the same time, it is becoming increasingly clear that water can play a crucial role in tackling issues such as climate change and subsidence as well as making agriculture more sustainable". The challenges ahead require continuous innovation. To make water management and governance future-proof, it is beneficial to look at fulfilling multiple objectives at once (win–win's or co-benefits) through a strategic approach to infrastructure refurbishment. In this way, costs and benefits are considered more holistically by learning from the past and from other cities that may experience similar challenges [3]. Several major investments are anticipated to be made to meet future challenges and to improve aging infrastructures.

## 3. Materials and Methods

### 3.1. Retrospective Analysis

The City Blueprint Approach (CBA) is applied to seven important periods in the history of Amsterdam starting from 1672. These seven periods are considered important distinctive stages in the development of the city's water management. The CBA has originally been developed by van Leeuwen et al. [19] and currently consists of three complementary frameworks (Figure 4). The main challenges of cities are assessed with the Trends and Pressures Framework (TPF). How cities are managing their water cycle is assessed with the City Blueprint Framework (CBF). Where cities can improve their water governance is assessed with the Governance Capacity Framework (GCF). An overview of the CBA is presented in Supplementary Materials I. In this research, we will use the TPF and CBF to perform a retrospective analysis. The CBA helps to gain a better understanding of the current status of water management and governance. It focusses on threats, weaknesses, strengths, and potential solutions. Applications of the CBA in the cities of Seoul, Cape Town, and Sabadell have been published in this journal, among other things, as contributions to a Special Issue on water management and governance in cities [20].

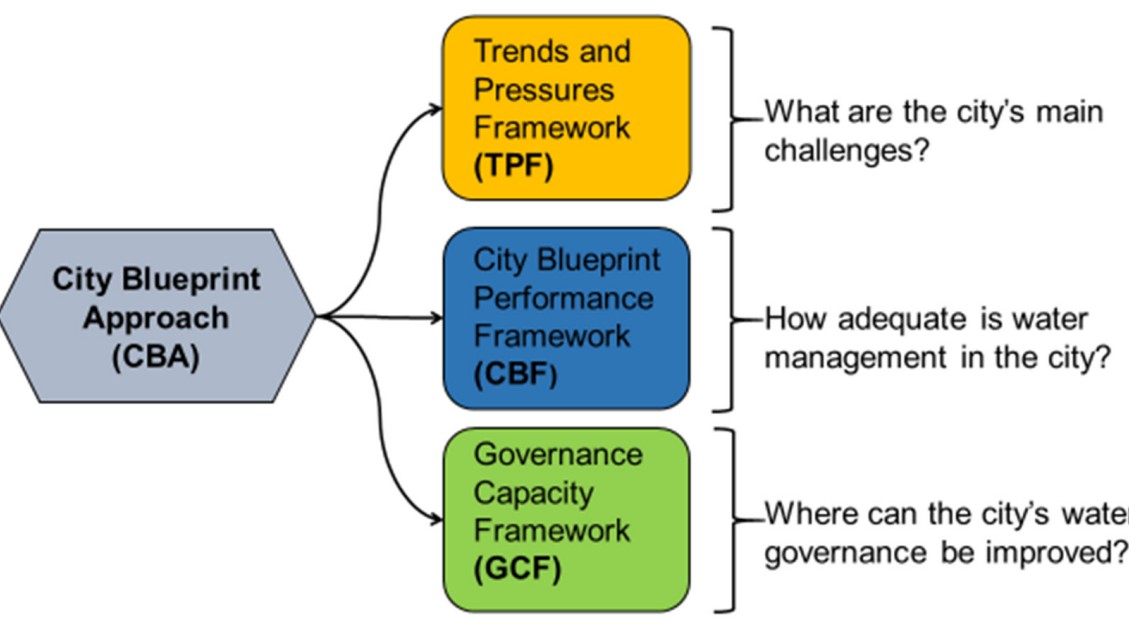

**Figure 4.** City Blueprint Approach [14–16].

The present situation of water management and governance in Amsterdam (2018–present) has been examined in previous research. This research analysed the other six periods. A map with the most important water-related connections of Amsterdam is provided in Figure 5.

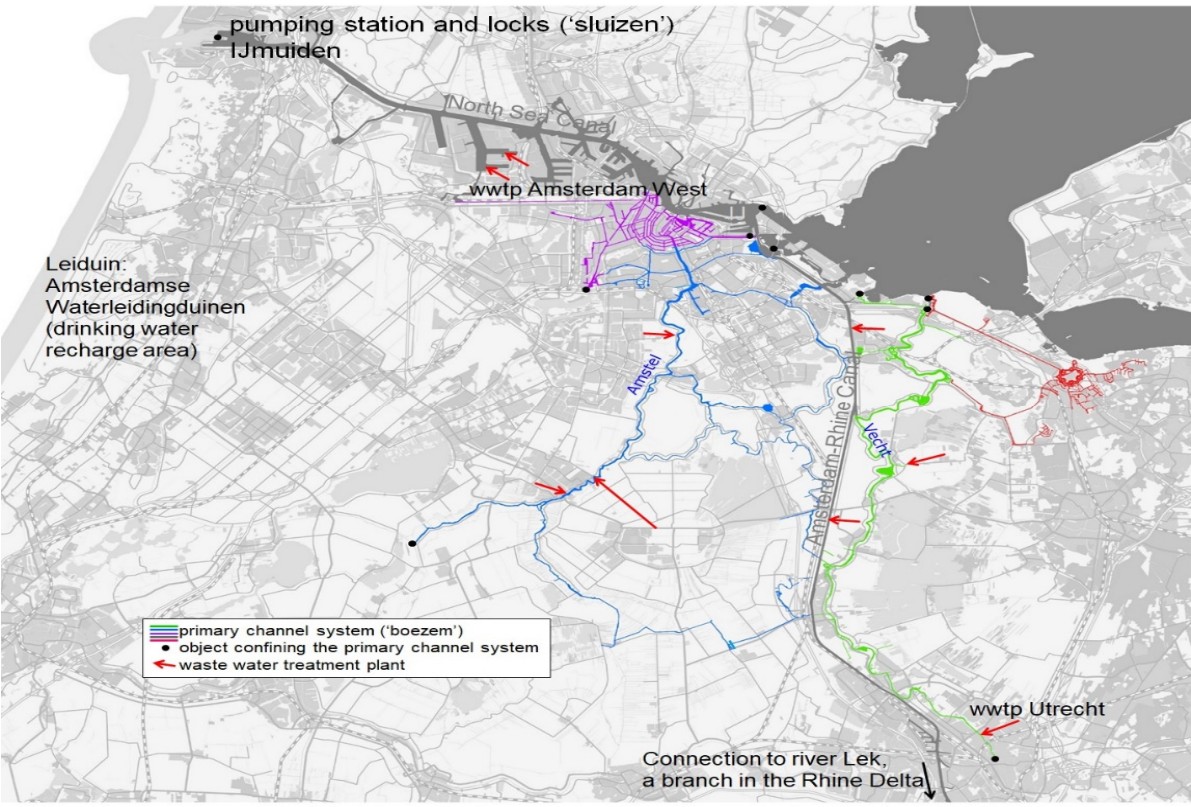

**Figure 5.** Geographic map showing the water-related connections of Amsterdam. The water intake is from the Amsterdam-Rhine Canal, about 10 km south of the city of Utrecht.

### 3.1.1. Trends and Pressures Framework (TPF)

The TPF consists of 24 indicators that are divided over social, environmental, financial, and governance categories. They are scored on a scale from 0 to 10, which is divided into ordinal classes as a degree of concern. Therefore, a low score represents a low challenge for IWRM. A revision in the TPF took place in 2020 owing to the inclusion of World Bank governance indicators and air pollution in the TPF. The Trends and Pressures Index (TPI) is the arithmetic mean of all TPF indicators. Details of the methodology, the scoring methods, and data sources have been provided in great detail [14].

In this research, we look back in time and score each indicator in the selected periods to see how these trends and pressures have evolved over time. Unfortunately, quantitative data are not available for many of these historic periods. Therefore, the scores were calculated with the use of the best historical data available. These data were obtained through literature reviews and expert interviews. Furthermore, in this study, we have added an indicator from an earlier version of the framework, namely the quality of surface water, because surface water quality plays a dominant role in the history of Amsterdam.

### 3.1.2. City Blueprint Framework (CBF)

The CBF consists of 24 indicators divided over seven main categories: (I) basic water services, (II) water quality, (III) wastewater treatment (WWT), (IV) water infrastructure, (V) municipal solid waste (MSW), (VI) climate robustness, and (VII) plans and actions [14]. The CBF provides an overview of a city or region's strong and weak points, which can be used for long-term strategic planning [3]. An overview of the indicators of the CBF is presented in Table 1.

The indicators are scored on a scale from 0 to 10. The lower the score, the more room there is to improve its performance. This study uses the revised version of August 2020 and, therefore, also provides for an update of the CBF of Amsterdam [15]. The City Blueprint analysis was also carried out over the various periods (Section 4) to illustrate the developments of water management and governance in Amsterdam.

### 3.2. Data Collection

To collect data for the CBF and TPF for the year 2020, the standardized questionnaires were used [14,15], whereas for the historic analysis, a literature study was performed, followed by interviews with experts. The literature study provided substantiation and data for the TPF and CBF. In the literature study, quantitative as well as qualitative historic data were collected. Historical quantitative data are not available for all indicators, hence this study relies partly on qualitative historical records. The data extracted from the literature search were supplemented and validated through expert judgment. In addition, the literature search was used for further contextualization and characterization.

**Table 1.** City Blueprint Framework [15]. WWT, wastewater treatment; IWRM, integrated water resources management; MSW, municipal solid waste.

| Goal | Baseline Performance Assessment of the State of IWRM | | |
|---|---|---|---|
| | (1) | Basic water services | 1. Access to drinking water<br>2. Access to sanitation<br>3. Drinking water quality |
| | (2) | Water quality | 4. Secondary WWT<br>5. Tertiary WWT<br>6. Groundwater quality |
| | (3) | Wastewater treatment | 7. Nutrient recovery<br>8. Energy recovery<br>9. Sewage sludge recycling<br>10. WWT energy efficiency |
| **Framework** | (4) | Water infrastructure | 11. Stormwater separation<br>12. Average age sewer<br>13. Water system leakages<br>14. Operation cost recovery |
| | (5) | Solid waste | 15. MSW$^2$ collected<br>16. MSW recycled<br>17. MSW energy recovered |
| | (6) | Climate adaptation | 18. Green space<br>19. Climate adaptation<br>20. Climate-robust buildings |
| | (7) | Plans and actions | 21. Management and action plans<br>22. Water efficiency measures<br>23. Drinking water consumption<br>24. Attractiveness |
| **Data** | Public data or data provided by the water and wastewater utilities | | |
| **Scores** | 0 (low performance) to 10 (high performance) | | |
| **Overall score** | Blue City Index®(BCI), the geometric mean of 24 indicators | | |

In addition to reviewing existing literature and reports, interviews with professionals on the various issues were conducted to gain a deeper understanding of how water management and governance developed in Amsterdam. The expert scores obtained through in-depth interviews were aggregated into the final scores for each of the seven historical periods. As such, various experts from the local water authority Waternet, but also from outside the organisation, were interviewed to provide the scores of the indicators and their perspectives and insights on the various themes. The interviews were semi-structured to ensure that all elements are taken into account. Each indicator with the corresponding method and score was explained to the interviewee. After that, the interviewees reported per period how the score for each indicator changed over time. The interviewees were also asked to explain why they assigned a certain score to an indicator. For the final scores, the average of all scores of the interviewees was calculated.

## 4. Results

In Figure 6, Amsterdam's water management performance (assessed by the City Blueprint methodology) is visualised over time. It shows how Amsterdam improved its water management and governance. It also shows that it took centuries to become water-wise. Amsterdam's overall water performance score, the BCI, has been very low, and gradually increased from 1.1 to 1.9 until 1950. In 1970, the city had a BCI of 4.3. In the last 50 years, the BCI increased to 8.7. The scores of the TPF indicators for different time periods are shown in Figure 7. Supplementary Materials II (pages 37–41) show each major trend and challenge over different time periods. The final scores of the CBF framework are also provided in Supplementary Information II. In the next sections, the results of the historic analyses of water management and governance in Amsterdam will be discussed in more detail.

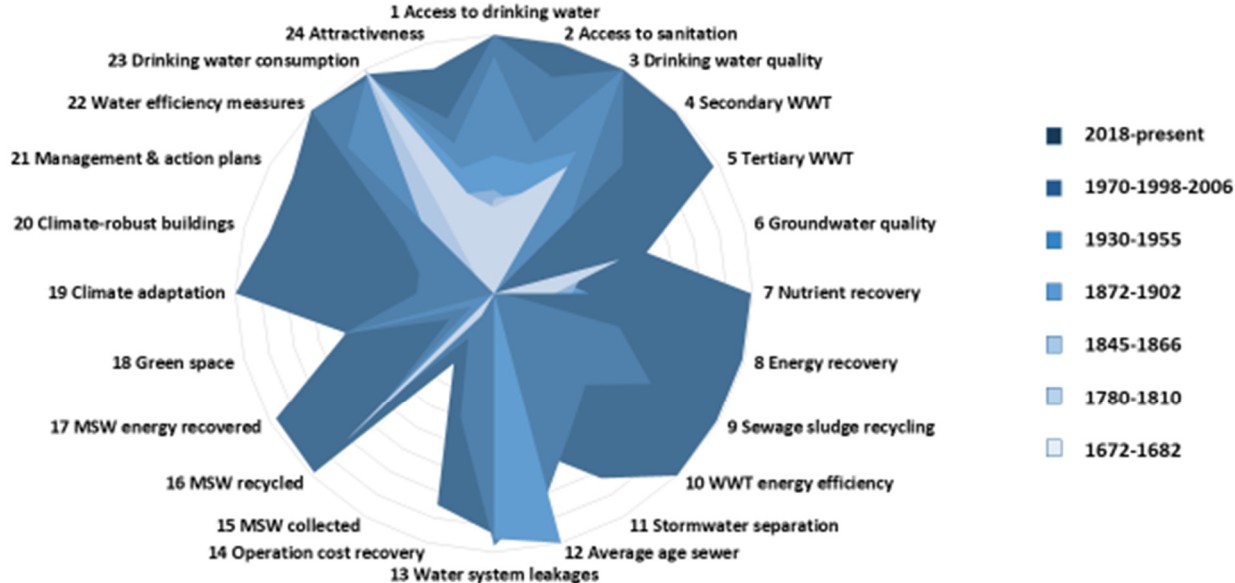

**Figure 6.** Retrospective analyses of Amsterdam using the CBF [15] over a period of 350 years.

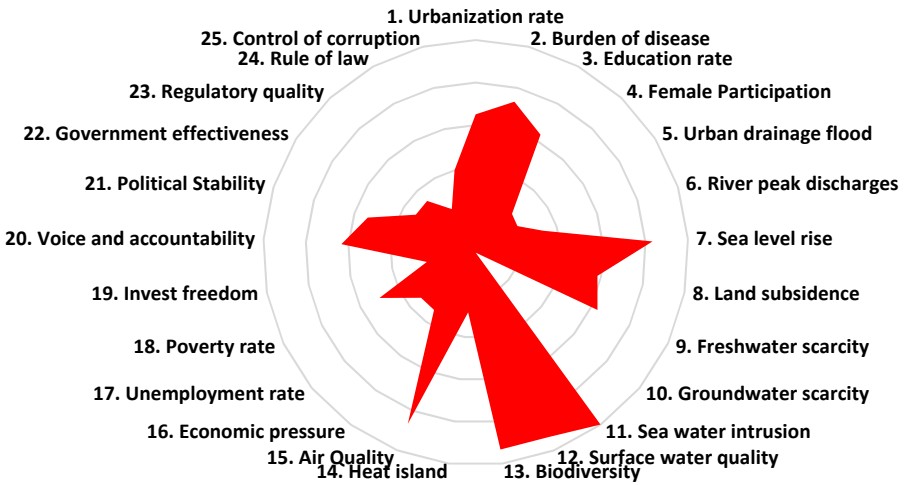

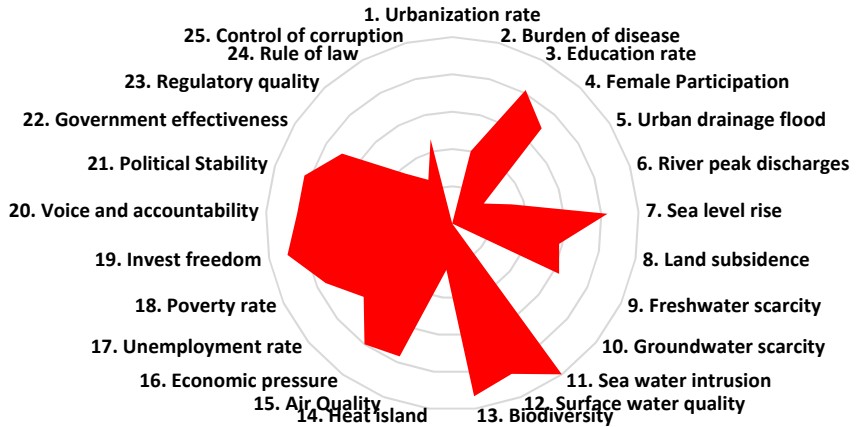

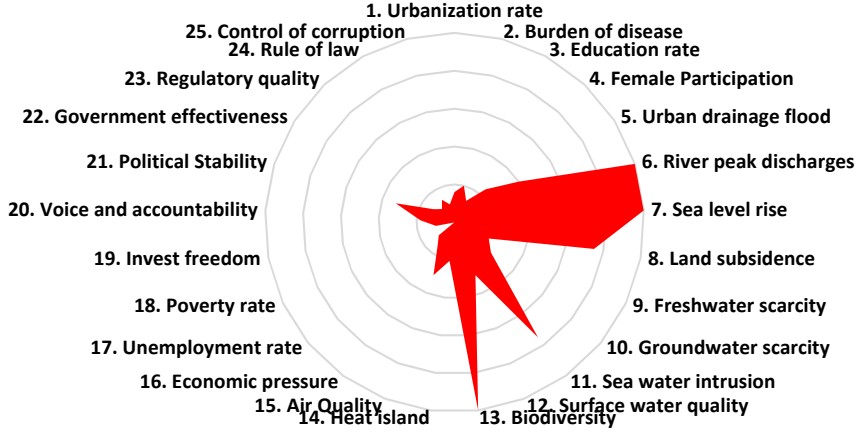

**Figure 7.** TPF of Amsterdam in 1672–1682 (**top**), 1780–1810 (**middle**), and 2018–present (**bottom**).

## 5. Discussion

### 5.1. Challenges Encountered by Amsterdam over the Past Centuries

From Figure 7 and TPF analyses for other periods (see Supplementary Information II, pages 37–41), it can be observed that the developments and trends in Amsterdam have evolved from crisis to crisis. For instance, over time, the various epidemic outbreaks have changed the burden of disease. Multiple epidemics have occurred, such as plague and

cholera. Furthermore, there are also interconnections. For example, the better the economic prosperity and employment rate, the more people settled in the city. The overall pattern is a decrease in pressures as many TPF indicators decreased over time owing to social developments and improved technologies and management practices. Some indicators continue to indicate a pressure (i.e., still showing high scores) owing to the geographic location of the city. The most remarkable result of the CBF analysis of Amsterdam is that the city had very low BCI scores for centuries. In 1970, the BCI score was 4.3 and only during the last 50 years has the BCI gradually increased to 8.7. The implication for other cities cannot be understated. The message is that it is feasible to improve IWRM in a couple of decades provided that there are clear plans, actions, and resources [3].

*5.2. Drinking Water and Health*

In the early years, citizens were able to drink surface water, i.e., from river Amstel and the city's canals, as they were not yet polluted and still contained an abundant fish population [21]. As a result of population growth, surface water became gradually more polluted. This is why breweries in the city switched their drinking water source and started using water from the Haarlemmer lake and river Vecht in the vicinity via boat transport [20]. This drinking water was not only used by the breweries, but also sold to citizens.

Unfortunately, only rich people could afford this water and poorer citizens were restricted to polluted drinking water from canals as well as rain barrels [22]. Drinking water supply by ships was hampered during the winter when the rivers were frozen [23]. An icebreaker pulled by horses had to clear the waterways, which was exhausting work and incurred high maintenance costs. Water shortage and great effort of icebreakers increased the price of drinking water and caused health issues in the city, predominantly among the poor people. It was Dr. Samuel Sarphati, a medical doctor and city planner in Amsterdam in the mid-19th century, who observed the poor hygienic conditions among the poor.

Plans were regularly made to search for drinking water in the deeper soil layers, but the drilling did not produce satisfactory results [22]. After a few harsh winters and problems with drinking water supply by ship, this resulted in the establishment of an organisation named 'Fresh Water Society' (in Dutch, 'Versch-Water Society') [23,24]. The public administration introduced all kinds of rules and regulations, and an agreement was made with the beer brewers. The 'Fresh Water Society' continued to fetch water from the river Vecht, and punctually implemented all regulations regarding the cleaning of the ships, and the regular water provision [24]. As larger ships could not bring water into the canals because of their size, the water was transferred to smaller ships that transported the water to the citizens. In 1789, the city council possessed underground drinking water cellars and tanks that were filled with water from the river Vecht to meet Amsterdam's growing water demand [21]. The need for supply of fresh water was fuelled by incursions by the Prussians (1787). In 1806, a meeting was held between the city council and the brewers and they agreed to install freshwater tanks [22]. Twelve freshwater tanks were allowed to be rented by the brewers so that they had a reserve stock in case daily supplies decreased. The drinking water status of Amsterdam before 1850 is similar to what we currently observe in a number of developing countries and illustrates the great efforts needed to secure water supply. Bringing good plans together, implementing them, and adequate funding proved to crucial in Amsterdam [22].

Compared with The Netherlands, France and England were far ahead in the development of drinking water infrastructure [22]. There were many initiatives, plans, and even organisations with different ideas about water supply. However, owing to lack of capital and/or cooperation with the city council, these were all nipped in the bud. The English and French were instrumental for Amsterdam. With the supply of financial and human capital in the form of English pipelines and machines and the perseverance of Jacob van Lennep (state attorney and poet) from Amsterdam, a drinking water pipe was constructed from the dunes at the coast to the city of Amsterdam [25]. Drinking water was obtained from the dunes where Jacob van Lennep, Ferdinand Huyck, and other noble lords owned

property. They founded the dune water company to supply drinking water through a pipe from the dunes to the city. The project was led by the English engineers John Aird, Charles Burn, and Bland William Crocker.

The day of 12 December 1853 was finally the day on which the first buckets of dune water could be tapped in Amsterdam (Willemspoort). Shortly afterwards, the dune water supply system expanded further within the city [25]. Houses in Amsterdam could subscribe to dune water pipe connection. In 1866, there were 56 taps in the city where buckets of water could be obtained. Some citizens were still concerned about the quality, but in 1854, some doctors declared that the drinking water was of sufficient quality to drink safely [22]. Until 1870, ships remained active in fetching water from the Vecht, but this gradually ended. Dune water was of great importance because of its hygiene, which was important owing to the fight against cholera in the preceding years. In 1866, a Christian foundation was set up to provide free water to poorer residents of the city, which contributed significantly to the preventive fight against cholera [21]. Because the demand for dune water continued to increase, an increase in the extraction and construction of a second pipeline from pumping station Leiduin to Amsterdam was necessary.

Dune water consumption increased strongly from 1853 onwards due to a rapid population increase. This caused problems with maintaining adequate water supply [25]. In 1885, the municipality granted the dune water company a new concession with the obligation to construct a pipeline for water supply from the river Vecht [25]. However, the water from the Vecht was not considered suitable drinking water because of the quality, which is why a double pipeline network was installed in the city. Water from the Vecht could not be supplied for domestic use, but was used for the fire brigade, flushing sewerage, and industry [25]. In the concession of 1885, the municipality stipulated that the dune water company had to transfer a large part of its income to the municipality [23]. This arrangement caused financial and technical problems for the company. The demand for dune water continued to increase, so water supply capacity had to follow. On 6 July 1889, an emergency measure was passed authorizing the dune water company to supply Vecht water in case of need for housing, but only for bath appliances, water boxes, and garden sprinklers. Since 1885, part of the municipal council wanted drinking water supply to become a municipal company. On 1 May 1896, the dune water company was taken over by the municipality of Amsterdam and its name was changed into municipal water pipes (in Dutch, 'Gemeentewaterleidingen') [25].

Further expansion was necessary owing to continued population growth. However, many ideas were rejected because of health and other risks [21]. As a result, the municipal water supplier was forced to significantly expand the dune water extraction at a location named 'Leiduin' and to construct new transport pipelines between this location and Amsterdam [22]. In 1916, large parts of Amsterdam's surroundings were ravaged by a major flood. As a result, the municipal supplier provided water to other cities for years [24,25]. Water supply increased by more intensive extraction of deep dune groundwater, as a result of which the deep freshwater resources were overexploited. In order to meet the demands, a start was made with water supply from other lakes (the Loosdrechtse plassen) in 1932 [26]. Ir. Bierman, who had studied Amsterdam's water supply during the Second World War, provided his 'Report 1948' to the municipal council [22]. A proposal was made to construct a pipeline between the Amsterdam-Rhine Canal and the dune water extraction to supplement the dunes with river water that could infiltrate and mitigate the over extracted groundwater resource. After the Second World War, water supply in Amsterdam was firmly established by implementing this plan [22]. The new plan included infiltration of the dunes and improvement and extension of the lake's water supply system. These plans were conceived and quickly implemented owing to the rapid growth of the city and its water consumption. For this plan, the municipality and the province had to collaborate, which is why, on 14 December 1950, a Regional Water Transportation Company (RWTC) was founded [25]. From 1957 onwards, the RWTC takes in water from the Amsterdam-Rhine Canal, i.e., the river Lek, which is a branch of the river Rhine in the Rhine delta (Figure 5).

Afterwards, this water is filtered and then transported to the dunes [21] and replenishes drinking water in the dunes.

In Leiduin, the capacity increased to 54 million m$^3$ per year through modernisation and expansion of the filter company [26]. In 1961, the old steam pumping station was decommissioned and a new pumping station with an electrically driven pump was put into use. In addition, a new transport line was built. Furthermore, in 1963, the International Rhine Commission was established in consultation with governments of various countries to clean the river Rhine and keep it healthy [21]. In the following years, the company's extraction capacity increased further by the construction of new infrastructure. Unfortunately, the river Rhine became increasingly polluted and saltier, partly due to intensive agriculture and industry (due to potassium mines upstream of the river Rhine). The various drinking water utilities that used Rhine water were joining forces and tried to tackle the problem both in The Netherlands and the foreign Rhine bordering countries [21]. In addition, the companies invested further in water treatment technology to supply drinking water of good quality.

Most countries add chlorine to the drinking water to control bacteria and viruses. However, the by-products of chlorine can be slightly carcinogenic [26]. This is why drinking water utility Gemeentewaterleidingen excluded chlorination. From 1983 onwards, the utility succeeded in providing reliable drinking water without adding chlorine. In times of emergency, however, there is always a chlorine dosage available [21]. Over the years, almost the entire dune area has become the property of the municipality of Amsterdam and turned into a protected water extraction area [26]. The area around Leiduin is used as a nature reserve, allowing recreational activities. In recent years, Waternet has been fully engaged in the integration of water extraction and nature management [26].

*5.3. Wastewater Disposal, Treatment, and Surface Water Quality*

Until the 19th century, it was common to dump excrements and dirt into the canals via gutters. As a results, the canals in Amsterdam served as open sewers [27] and water quality in the canals was very poor. As early as 1481, there were complaints in the city about the dirt and the stench. Despite the availability of partitions under the bridges, the situation did not improve. Owing to the open sewage system, diseases spread quickly and the odour nuisance was omnipresent [27]. The canals were flushed by the tidal flows that flushed the sewage out of the city [27,28]. Canals that were not flushed properly were dredged or were filled up [21]. This eased up traffic and improved the air quality too. Complaints about odour nuisance increased in the 19th century. In summers with lower water levels, the odour nuisance was unbearable. In other European cities, sewers were already installed at that time (e.g., London in 1840, Hamburg in 1842, and Paris in 1850), but Amsterdam lagged behind [21]. There was also a gradual understanding that poor hygienic conditions could be the cause of recurring epidemics of infectious diseases (such as diphtheria, tuberculosis, typhus, measles, red spark, and malaria). Infant mortality was high and the city was often ravaged by cholera or smallpox outbreaks. Furthermore, the industrial revolution exacerbated the problem with the discharge of wastewater from factories into the surface waters [21,27].

The wastewater system improved after the establishment of the Public Works Department (named 'Dienst der Publieke Werken') in 1850. Wastewater treatment became even more necessary thanks to the planned building of the North Sea Canal that would also shut off the canal system, affecting the natural flushing. Moreover, the cholera epidemic of 1866 initiated discussions in the municipality about a sewerage system as a solution had to be found for the discharge of wastewater [21]. In 1870, the Liernur sewer system came into operation [28]. Before the municipality started to apply this system, this initiative was already carried out on a small scale by private individuals. This system was the first large-scale sewerage system of the city. Wastewater was collected at a central location, after which the faeces could be used as fertiliser for agriculture. In 1912, the Liernur sewage system was discontinued because it could not process the discharge of rainwater and

domestic water [28]. Water use increased owing to the construction of the water supply system. This made the faeces too liquid, causing the system to function insufficiently.

In 1906, it was decided to build a mixed sewer system outside the city [29]. The new main sewer brought in water under the influence of gravity. The sewage system ended at a pumping station at Zeeburgerdijk, after which it was pumped to the inland sea (which became a lake after the completion of the sea barrier in 1932) [28]. A mixed sewer system was chosen for hygienic reasons, but most of all, because of the lower cost compared with a separate sewer system [28]. The collected wastewater was discharged into the lake IJssel without pre-treatment. Overflow of the sewage system was still discharged into the canals and the city centre because it was directly connected to the canals. The volume of wastewater fluctuated strongly as a result of changing population size and prosperity [28]. In the 1930s, the 16th century city centre was connected to the sewer system and more wastewater had to be processed. The construction of sewerage in the city centre was co-financed by the government work fund as an attempt to combat unemployment during the global economic crisis. In 1926, the first large-scale wastewater treatment took place because the waste water treatment plant (WWTP) in Amsterdam western area became operational. In addition, stormwater discharge took place by means of a separate sewer system. Later, another five WWTPs were constructed because of the large amount of wastewater produced by the fast-growing city [28].

After the Second World War, major changes took place. Not only was the quality of the sewage system assessed on the basis of its importance for public health, but the environment also received increased attention. During this period, many technical developments were implemented for wastewater collection systems and improvement of the quality of the water. This new vision on the quality of waste and surface water further increased investments in the sewer system. Citizens agreed to spend their tax money on improving the wastewater system. Moreover, the municipality was forced to invest owing to the tightening of national environmental standards. In 1982, a large WWTP (650,000 inhabitant equivalent) became operational, ending the discharge of untreated wastewater [28].

By 2005, a new project started as the WWTPs in the east and the south no longer complied with the requirements of Dutch and European legislation and regulations [30]. The regional water authority Amstel, Gooi, and Vecht (AGV) and the municipality decided to build a new centralised WWTP in the Amsterdam West Port Area [31]. The new WWTP applied advanced technologies, resulting in high-quality effluents water. More specifically, the treatment process removed nutrients at minimal chemical input and energy use. Furthermore, the energy content of the sludge and biogas was utilised by the Waste and Energy Enterprise (WEE) or AEB of the city [31]. This project was one of the largest infrastructure projects in Amsterdam in recent decades. Moving WWTP from south and east to west, downstream of the river Amstel and the canal zone, improved water quality substantially. Another project that has ensured that virtually no untreated wastewater would end up in the canals is the project 'Schoon Schip' (i.e., clean ship). In collaboration with the municipality and the Dutch Ministry of Infrastructure and Water Management, the water utility of Amsterdam has provided connections of all houseboats to the sewerage system [32]. As a result, water quality in the river Amstel and in the canal zone meets high-quality standards as applied in official swimming locations most of the time. Only after sewage overflow, which occurs on average five times per year, water quality is unsuitable for swimming for three to five days after an overflow. Future challenges concern improving ecological water quality and dealing with the management of pressures such as boat traffic and structural diversity.

*5.4. Solid Waste*

Throughout the centuries, waste processing in Amsterdam followed a pattern of outsourcing and 'do it yourself' [31]. In 1673, the management of solid waste passed to the regents of the poor chapel orphanage, a semi-governmental institution. The orphanage could use an extra source of income because—due to the plague epidemic—the organisation

was required to take in many more orphans. The municipality tried to create a win–win by lowering the subsidy to the orphanage and cleaning the streets, but the orphanage could barely cope with the waste collection tasks. As a consequence, dirt remained on the street, residents complained, shortages ran up, and reorganisations followed [33]. In 1804, the municipality outsourced the solid waste service to Nicolaus Sieburg and Martinus van der Aa [33]. For 40 years, they kept the streets clean. However, little profit was made, causing the company to go bankrupt. In 1848, the contract to collect waste was taken over by the Association for Agriculture and Land Development, run by Dr. Samuel Sarphati [33]. Sarphati's goal was three-fold: (i) to promote public health, (ii) to improve agricultural land, and (iii) to create employment. In 1850, Sarphati's second initiative was launched. He set up a street sweeper service to reduce pollution in the streets [34].

The association did not last long, because the municipality took matters into its own hands again. A private urban cleaning service named 'Dienst der Stadsreiniging' was established in 1877 [32,34]. The amount of solid waste accelerated due to rapid population growth. On 21 May 1913, the city council decided that municipal waste needed to be processed in a waste incinerator to be built in the north of Amsterdam. The purpose of the incinerator was not only to get rid of the waste, but also to produce electricity by converting heat into electricity [34]. Five years later, the incinerator was commissioned with some delay as a result of the First World War. In 1919, the city's first incineration plant was put into operation. The released energy could be used and the residual waste was used as building material [34]. In 1969, the old installation was replaced by a new installation with more combustion capacity and, therefore, greater efficiency.

In 2001, the city established six waste-collecting locations where bulky household waste, hazardous waste, and electrical devices were collected [34]. These sites were used to increase the reuse of bulky waste to a ratio of 70% [34]. Waste was no longer considered to be an annoying side effect of urban life, but as a profitable source for recycling of useful materials and 'clean' energy generation [33]. In Amsterdam, the Municipal Waste Management Service became the WEE in 2003. In 2014, the organisation became private in order to develop as 'the producer of sustainable energy in Amsterdam' [33]. In the early days, almost everything was recycled, but this decreased owing to the increase of the share of non-recyclable materials. In recent years, the WEE has managed to increase the recycling rates again.

Moreover, the WEE expanded and opened new furnaces and even started to import waste from abroad; one-sixth of all Dutch waste was processed in Amsterdam [33]. In 2006, WEE entered into a partnership with Waternet, providing mutual synergies, i.e., (a) the residual heat from the combustion gases is used to make the treatment process more effective; (b) the sewage treatment runs on the electricity generated by the waste incineration; and (c) the energetic yield of biogas, which is released during the purification of the sludge, increased by one-third [34].

### 5.5. Green Space and Climate Adaptation

Throughout history, Amsterdam never had a low proportion of green and blue space. In the VOC (United East Indies Company) and French time (i.e., early 17th century to early 19th century), Amsterdam scored well on greenery in the city because the area cleared for city expansion was not yet fully built up. During the industrial period, the population increased, causing the city to expand and to become more densely populated. This caused a decrease in the share of green space in the city. However, in 1864, a public park (Vondelpark) was built in the middle of the expanding city that provided citizens a place to enjoy nature [24]. The park was used intensively, causing the water quality to deteriorate [20]. In 1901, Jacobus Pieter Thijsse suggested to create a forest close to Amsterdam. This forest (named Amsterdamse Bos) was created between 1927 and 1964. In early years, its creation was a project to address employment during the global economic crisis.

The percentage of blue space in the city is high owing to the city's canal system. From an historical perspective, the abundance of water was not only for the benefit of transport

and drainage, but it also had great aesthetic value. Ever since the 17th century, citizens preferred to live near water and make frequent use of it [20]. The proportion of green space in the city can still be improved and great attention is paid to this in recent years. Today, climate change is an important driver of water management and governance in cities. Changes such as drought, heat, and more frequent and extreme downpours have a major impact on urban water management. Therefore, the government adopted the national adaptation strategy for climate change in 2007. In 2014, the water utility of Amsterdam started a programme in collaboration with the municipality: Amsterdam Rainproof. The aim of the programme is to make Amsterdam rainproof by 2050.

### 5.6. Planning and Operations

Ever since there was civilization in Amsterdam, there has been an administrative organisation for water. It started with dike management and later also water level management. Regional water authorities were established step by step. At the end of the 17th century, water management in Amsterdam was well organised. This was mainly thanks to Mayor Joan Hudde. In the national disaster year of 1672, Hudde was appointed mayor and was reappointed 27 times between that year and 1703 [22]. During this period, the city council took many decisions to manage the water system, such as the building of sluices (e.g., Hogesluis and the Amstelsluizen), water mills, and various bridges [22]. The construction of sluices made it possible to better regulate the water levels in the canals. Clean water was taken in from the lake IJ at high tides. Dirty city water was drained at low tide. During all these periods, water tasks were municipal services [21].

In 1970, the Pollution of the Surface Water Act came into force, making the purification of wastewater a regional responsibility. As a result, the regional water authority Zuiveringsschap Amstel- en Gooiland (AGV; 1973) was established as well as the Amstel and Vecht regional water authority for the province of North Holland [22]. The municipality retained its active purification task and the ability to collect pollution tax. In 1990, new legislation was introduced. The Water Management Act further delineated the water tasks. The city handed over its water management and wastewater treatment tasks to the AGV. After much consultation, the Water Management and Sewerage Service (WMSS) was established, which included the tasks of both municipality and the regional public water authority Amstel, Gooi, and Vecht [21,35]. In 2006, Amsterdam decided to transfer the drinking water tasks to AGV, which resulted in the establishment of Waternet. Waternet was founded by the regional water authority and the municipality and carries out water-related tasks in an integral manner. In other cities in the Netherlands and beyond, all tasks are assigned to different organisations. Since the establishment of Waternet, water management improved substantially, demonstrating the effectiveness of an integral approach to water management and governance.

### 5.7. The Application of the CBF for Historic Analyses

Amsterdam has seen many developments in the field of water management and governance. The City Blueprint (Figure 6) shows at a glance how water management has changed over time. The TPF applied in different periods (Figure 7) shows which pressures dominated and, based on the CBF, which subsequent actions were taken to reduce these pressures.

The analytical framework was useful for assessing water management and governance in different historical periods. This comprehensive framework [14–16] has been applied to assess water management of 125 cities in 53 countries in order to provide a frame of reference to enhance city-to-city learning. The historical analysis of Amsterdam's water management demonstrates the applicability of the City Blueprint as a frame of reference to also learn from the past. Despite the complexity of the urban water system and limitations in the provision of historical information, this study demonstrates that it is possible to apply the framework and develop meaningful insights.

The connectivity between the indicators of both the TPF and the CBF is important to consider because there is often a clear link between pressures and measures. The framework approach is based on contemporary times and the methodology is not tailored to assess the indicators in other time periods as accurately as today. For example, the perceptions on drinking water quality have changed over time. Water that people drank in previous centuries will not meet the recent quality requirements, but for the people who lived in these past centuries, drinking water quality may have been quite sufficient. Another example is the poverty rate. Poverty is defined as the percentage of the population living below the poverty line of 1.9$ per day. The methodology is correct for the present time, but not applicable for other periods. Therefore, the scores of the TPF and CBF in this study are based on expert judgment that account for the historical context.

For this study, multiple interviews were conducted to score the indicators of the TPF and CBF in the different historic periods. Stakeholders who were interviewed are leading experts about the historical events. In addition, the scores were checked and justified by means of literature research. Furthermore, when completing the scores, it is important that the interviewees have a good overview of the events and knowledge of all areas of water management and governance. That is why only people who meet these knowledge requirements were interviewed.

The holistic overview of the historic developments in water management and the causes for that provide relevant lessons. One of these lessons is that Amsterdam's water management performance accelerated after each consecutive crisis, which can be described as a pattern of problem-shifting. Problem-shifting refers to a process where short-term and reactive management solutions in turn create new water-related problems [36]. The second lesson from the study of Amsterdam is that cities can rapidly improve their water management in only a couple of decades (Figure 6).

It is interesting to explore if such an achievement is feasible for other cities that follow a similar trajectory. It should be realised that cities often face different pressures, causing these cities to make different decisions and follow different paths. To get a complete picture of which decisions are strategic in which situations, further research is recommended in other cities to better understand the role of contextual factors [37]. At present, a cluster analysis is carried out for the 125 cities that we have assessed and a clear process of problem-shifting can be observed. It is thus recommended to adapt the CBA method to make it more suitable for assessing the indicators in an historical context.

## 6. Conclusions

The lessons from Amsterdam are lessons of humility, of trial and error, and of becoming wise through injury and failure. Amsterdam may now be one of the best cities in terms of water management and governance, but it is not the result of a few years' effort. It took over six centuries to gradually develop and improve the water system (Supplementary Information III). In the last 50 years, the city rapidly improved its water management to become a water-wise city (Figure 6). Amsterdam faced many challenges that had their effect on society. These challenges impacted the development of water management and governance and shaped the city's water infrastructures. This research showed that especially the social and economic indicators were driving forces. The city council and the government have not been the limiting factor during these periods owing to the relatively high stability of government. Environmental indicators such as seawater intrusion and land subsidence have shaped Dutch regional water management for centuries. Climate change will bring this challenge to a new level, urging water management to creatively use the existing water infrastructure and improve it effectively. In addition, water availability and water pollution still are important challenges. However, for the city of Amsterdam, the spikes in the burden of disease caused by accelerated environmental pollution during periods of urban expansion necessitated public water infrastructure refurbishments.

Looking at how and when water infrastructure was built in the city, it can be observed that it often was the result of a crisis spurring international knowledge transfer and

practices. It is difficult to predict and avert a crisis in advance and often short-term strategies prevail over long-term strategies. Looking at Amsterdam's historical water management, a crisis typically resulted in a sense of urgency that was necessary for change. However, preventive measures and initiatives are more effective in reducing pressures on the city, social-technical system, and society as a whole. This is an important notion facing the challenges of climate change. Amsterdam's cholera epidemic in 1866 exemplifies this necessity for preventive measures. Owing to the construction of a clean drinking water supply network (i.e., a preventative measure), Amsterdam had fewer deaths from the cholera epidemic than other cities.

Cooperation and knowledge transfer from other cities and abroad are important too. This can be seen, for example, in the construction of the drinking water supply system in Amsterdam. The United Kingdom has played a major role in this, not only by knowledge transfer, but also financially. The drinking water system was developed under the guidance of private individuals and Englishmen. Many of the initiatives to solve challenges in the city started with citizens' initiatives and private individuals. Financial capital, knowledge, and good mutual cooperation are essential in the construction and implementation of water infrastructure and management. The lessons from this analysis can be best summarised by the seven C's of water-wise cities [3,34]:

1. Citizen-centred: create healthy and liveable cities for people.
2. Children and grandchildren first. There is a need for long-term strategies to provide intergenerational justice.
3. Co-creation: involve stakeholders right from the start.
4. Co-design: comprehensive and coherent planning by integrating water and other sectorial agendas.
5. Co-benefits or win–win's need to be explored, as this will lead to the following:
6. Cost-effective and efficient solutions that need to be shared by the following:
7. Collaborative learning, i.e., city-to-city learning, to speed up the process.

**Supplementary Materials:** I. General information about the City Blueprint Approach: https://www.ipr.northwestern.edu/our-work/research-tools-apps/water-insecurity/ (accessed on 30 March 2021). II. Master thesis report: https://library.kwrwater.nl/publication/62028203/ (accessed on 30 March 2021). III. Master thesis presentation: https://library.kwrwater.nl/publication/62028202/ (accessed on 30 March 2021).

**Author Contributions:** Research idea, K.v.L.; project management, K.v.d.L.; methodology, S.K., K.v.L., and S.P.; data collection, S.P. and M.O.; analysis, S.P.; investigation, S.P. and M.O.; writing—original draft preparation, S.P.; writing—review and editing, S.P., M.O., S.K., and K.v.L.; supervision, M.O. and K.v.L. All authors have read and agreed to the published version of the manuscript.

**Funding:** This research received no external funding.

**Data Availability Statement:** All data from this study are provided in the master thesis report provided in the Supplementary Materials.

**Acknowledgments:** The authors would like to thank the interviewees involved in the research who have given us valuable information. This paper would not have been published without the support and help of Kees Hogenes, Wilko Koning, and other colleagues of Waternet.

**Conflicts of Interest:** The authors declare no conflict of interest.

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
