# Peer review of "Retrospective Analysis of Water Management in Amsterdam, The Netherlands"

_water, doi:10.3390/w13081099_

Round 1
Reviewer 1 Report
The authors present a compilation of the work carried out using the CBA approach for different historical periods in water management in the Amsterdam region. Much of the work is related to the actions carried out by the Waternet company. The CBA approach is based on the application of three different frameworks: the TPF (Trends Pressures Framework), the CBF (City Blueprint Framework) and the GCF (Governance Capacity Frameworks). Each of these frameworks contains the definition of its own indicators.
The result obtained by the authors allows seeing how the CBA approach has made it possible to improve water management in Amsterdam. The objective pursued in the work is to show how the experience developed in this case can serve as a guide for its application in other parts of the Netherlands or the rest of the world.
From the formal point of view the document is correctly written. The explanations are easy to follow and the concepts presented are clear. There are some formal aspects to correct, mainly related to the management of references. Likewise, the characteristic structure of a research work is not seen but rather the structure of a report.
From the point of view of its content, the work presents an absolutely interesting subject, such as the problem of global water management in urban environments. However, as indicated above, the document does not have a clear scientific structure. A state of the art is not distinguished in the introduction, the gap in knowledge that the work intends to cover is not specified, and the objectives and novelty of the work are not explicitly indicated. In the materials and methods part, a large number of previous works are described, but the specific works and developments of this work are barely described. The results section interchangeably mixes the results with comments and discussions about them. In this section it is not clear how the data are obtained and how it has been processed. Finally, the Discussion section is very brief and does not delve into the analysis of the results. Perhaps a part of what is exposed in the Results section should be in Discussion.
In short, the article is easy to read, it contains information that may be interesting, but in my opinion it does not have the structure of a scientific work. Therefore, my recommendation is that the authors perform major revisions on their document before it can be accepted for publication.
Given that I consider the content and experience presented by the authors is very interesting, here are some of the comments that should specifically be taken into account to improve the document:
- References should be checked. There are too many references to documents, reports and websites that contain little scientific content. References to more scientific works (papers, books, ...) are missing. Specifically, it is curious that there is not single reference to the journal in which the publication of the work is proposed. If in a magazine that publishes hundreds of works per year there is none that can be mentioned, why would this be the most appropriate journal for the publication of the work? I believe that the authors could find in the journal enough references to similar and closely related works that would better mark the field of study.
- There are numerous references (22 to 31 and 33 to 37) in Dutch. From my point of view, the references should be mostly in English so that the scientific community can review them. In any case, if the circumstances require it and it is necessary to include the reference, in my opinion, it should be indicated (In Dutch) next to the reference.
- From my point of view, some references could be avoided as they refer to very basic or unnecessary information. Reference 4 refers to Sustainable Development Goals. The reference to the website does not add anything, since it is a widely known information in the scientific community. Reference 13 only presents the study area of the Waternet company. Perhaps this could be included in the text with a figure and no such reference would be necessary. Some references are not justified (34 and 36). In my opinion, the information they provide is not relevant.
- Some references contain links to websites. Some of these links do not work (references 7, 14, 15).
- Lines 70-72: the authors speak about previous works. What difference do these works have with the one presented in the document? Improvements and new features in the submitted work should clearly stand out from previous work.
- Figure 2. This figure is extracted from a previous work. My impression is that it is not clear from the text whether this figure is the current situation or the result of previous studies. In other words, are Figures 2 and 3 part of the background or the results that are presented in the work?
- After reading the introduction, my impression is several things should be clarified: the knowledge gap the work aims to cover, its objectives and contributions regarding the state of the art. In my opinion this is not clearly stated. Authors should work on this part to highlight these aspects.
- Personally, I think that the use of the Waternet brand results in some points excessive. As a researcher, I fully understand the need to frame the work in the context of the company in which it was developed. However, with the utmost respect, this continuous repetition of the company name use makes the document look more like a report, leaflet or on booklet than a scientific article.
- Line 180: Obtaining the data is indicated in a very generic way. Authors should provide more details about the origin of the data, its quality, and how they were structured. Likewise, the authors should emphasize the criteria used to analyze the opinion of the experts. I would dare to suggest that improving this aspect would be critical to the job. Perhaps one of the most significant contributions of the work has been the ability to apply the CBA over time.
- Line 202: Results section should be limited to the presentation of the study results only. From my point of view there are numerous aspects in this section that could be moved to the Discussion section. At the same time the Discussion is too short section.
- Since the results of the study are based on interviews, the contents of these interviews, their results and the way of analyzing to draw the conclusions are important. It is perhaps one of the main aspects of the job. Authors should give details about them.
- Line 544: I think the authors should justify the standardization method. What influence could other methods have on the results? I suggested the authors to discuss this influence.
- Line 545: The questionnaire in reference 11 seems important. However, in reference 11 the questionnaire does not appear. In this reference there is a link where the questionnaire can be viewed, but this link does not work. If much of the study is based on this questionnaire, could the authors include both the questionnaire and its results as Supplementary Material?
Additionally, I have detected some minor changes:
- The way words are divided at the end of lines is strange in some cases. Perhaps the authors should revise this way of dividing the words and instead use the justification of the text.
- Reference 7 is written in a different text format than the rest.
- Several consecutive blank spaces appear on some lines: lines 480 and 519
Author Response
Thanks for your comments. Our answers are in bold.
From the point of view of its content, the work presents an absolutely interesting subject, such as the problem of global water management in urban environments. However, as indicated above, the document does not have a clear scientific structure. A state of the art is not distinguished in the introduction, the gap in knowledge that the work intends to cover is not specified, and the objectives and novelty of the work are not explicitly indicated. In the materials and methods part, a large number of previous works are described, but the specific works and developments of this work are barely described. The results section interchangeably mixes the results with comments and discussions about them. In this section it is not clear how the data are obtained and how it has been processed. Finally, the Discussion section is very brief and does not delve into the analysis of the results. Perhaps a part of what is exposed in the Results section should be in Discussion. Thanks. We have addressed all these issues and revised the structure of the paper too.
In short, the article is easy to read, it contains information that may be interesting, but in my opinion it does not have the structure of a scientific work. Therefore, my recommendation is that the authors perform major revisions on their document before it can be accepted for publication.
Given that I consider the content and experience presented by the authors is very interesting, here are some of the comments that should specifically be taken into account to improve the document:
- References should be checked. There are too many references to documents, reports and websites that contain little scientific content. References to more scientific works (papers, books, ...) are missing. Specifically, it is curious that there is not single reference to the journal in which the publication of the work is proposed. If in a magazine that publishes hundreds of works per year there is none that can be mentioned, why would this be the most appropriate journal for the publication of the work? I believe that the authors could find in the journal enough references to similar and closely related works that would better mark the field of study. Reply: We are sorry for this. This is a good point. We have published some papers in Water, and edited a special issue on the challenges of water management and governance in cities in this journal in 2019. We have included this reference to this special issue.
- There are numerous references (22 to 31 and 33 to 37) in Dutch. From my point of view, the references should be mostly in English so that the scientific community can review them. In any case, if the circumstances require it and it is necessary to include the reference, in my opinion, it should be indicated (In Dutch) next to the reference. For the sake of transparency and traceability, we would like to keep these Dutch references in the paper as there are no English versions of this historic material. We have included (in Dutch) in all these cases.
- From my point of view, some references could be avoided as they refer to very basic or unnecessary information. Reference 4 refers to Sustainable Development Goals. The reference to the website does not add anything, since it is a widely known information in the scientific community. We have deleted this reference 4 and replaced it by a relevant UNEP report. Reference 13 only presents the study area of the Waternet company. Perhaps this could be included in the text with a figure and no such reference would be necessary. Thanks we have deleted this reference 13 as it is not really necessary for this paper. Some references are not justified (34 and 36). Reference 34 is very relevant for the solid waste challenges of Amsterdam and we like to keep it in the paper. In my opinion, the information they provide is not relevant. We have deleted reference 36.
- Some references contain links to websites. Some of these links do not work (references 7, 14, 15). Comment. Thanks for this. We have updated the links for reference 7 and 15. We have deleted reference 14
- Lines 70-72: the authors speak about previous works. What difference do these works have with the one presented in the document? Improvements and new features in the submitted work should clearly stand out from previous work. Thanks. We have addressed this in the last section just before materials and methods.
- Figure 2. This figure is extracted from a previous work. My impression is that it is not clear from the text whether this figure is the current situation or the result of previous studies. In other words, are Figures 2 and 3 part of the background or the results that are presented in the work? Correct. Yes we have analysed the city of Amsterdam three times. Figure 2 and 3 represent the TPF and CBF analyses done in 2020, so these are the most recent results applying the new CBF and TPF approaches for which the methodological references are included in the reference list of this paper.
- After reading the introduction, my impression is several things should be clarified: the knowledge gap the work aims to cover, its objectives and contributions regarding the state of the art. In my opinion this is not clearly stated. Authors should work on this part to highlight these aspects. Thanks, we have clarified this now in the last section before Materials and methods. NB we are not aware of any study of this kind,
- Personally, I think that the use of the Waternet brand results in some points excessive. As a researcher, I fully understand the need to frame the work in the context of the company in which it was developed. However, with the utmost respect, this continuous repetition of the company name use makes the document look more like a report, leaflet or on booklet than a scientific article. We see your point and we have deleted the name about 5 times and replaced it by water utility of Amsterdam
- Line 180: Obtaining the data is indicated in a very generic way. Authors should provide more details about the origin of the data, its quality, and how they were structured. Likewise, the authors should emphasize the criteria used to analyze the opinion of the experts. I would dare to suggest that improving this aspect would be critical to the job. Perhaps one of the most significant contributions of the work has been the ability to apply the CBA over time. We have clarified section 2.2 and specified the data collection process. We have not used specific criteria for the opinions. What has been done is to find the few knowledgeable experts still around that could be interviewed about the history of drinking water, surface water, waste water and solid waste. The views expressed by the experts confirmed the results from the historic literature.
- Line 202: Results section should be limited to the presentation of the study results only. From my point of view there are numerous aspects in this section that could be moved to the Discussion section. At the same time the Discussion is too short section. We have shortened the result section and moved all other text to the discussion and rearranged the numbering too. We kept structure of the topics
- Since the results of the study are based on interviews, the contents of these interviews, their results and the way of analyzing to draw the conclusions are important. It is perhaps one of the main aspects of the job. Authors should give details about them. See point 9
- Line 544: I think the authors should justify the standardization method. What influence could other methods have on the results? I suggested the authors to discuss this influence. See point 9.
- Line 545: The questionnaire in reference 11 seems important. However, in reference 11 the questionnaire does not appear. In this reference there is a link where the questionnaire can be viewed, but this link does not work. If much of the study is based on this questionnaire, could the authors include both the questionnaire and its results as Supplementary Material? We have decided to include the three frameworks of the CBF as reports with links in order to reply to detailed questions about the methodology. Perhaps it is your browser. In the document the link works and then you can download the questionnaire as pdf. The info of the frameworks is also given in the supplementary information: https://www.ipr.northwestern.edu/our-work/research-tools-apps/water-insecurity/.
Additionally, I have detected some minor changes:
- The way words are divided at the end of lines is strange in some cases. Perhaps the authors should revise this way of dividing the words and instead use the justification of the text. We are not aware of this, and perhaps this should be changed during the production process by the editor of the journal
- Reference 7 is written in a different text format than the rest. Revised
- Several consecutive blank spaces appear on some lines: lines 480 and 519. We have updated the figures and the text as good as we can

Reviewer 2 Report
The article is well written and structured. It presents very interesting and useful information. I only want to highlight two points:
The representation of the value in Table 2 with a line graph (time line in x-axis) would help to picture the evolution of the index in a better way.
In my opinion, the conclusion is a bit too broad. My recommendation is to try to develop it in a more simplified way.
Author Response
The article is well written and structured. It presents very interesting and useful information. I only want to highlight two points:
Thanks for your comments. Our replies are in bold
The representation of the value in Table 2 with a line graph (time line in x-axis) would help to picture the evolution of the index in a better way. We changed the table into one figure and referred to the supplementary information for the other time periods.
In my opinion, the conclusion is a bit too broad. My recommendation is to try to develop it in a more simplified way. We have shortened the conclusions
Reviewer 3 Report
The authors focused on an interesting topic. The topic is related to the journal, and it is updated given the EU policies. Nonetheless, some changes are required to maximise the quality of the manuscript. Most of the challenges are aimed to add more information or modifying the existing information into a new format of sections to facilitate the readers with an overview of the ideas presented in this paper. Following, I include some comments:
1- I suggest changing Figure 1 with a new Figure that shows the city as a map. If possible, authors have to generate it. In this map, authors can indicate the type of land use (green areas/residential areas/industrial areas…). Having an image of the city itself does not add much value to the paper, especially if the picture does not belong to the authors.
2- The introduction should be reworked. Authors must differentiate the problem statement from the related work. In the introduction, authors have to define the problem they are facing and their objective. The content linked to the related work has to be described in a related work section. This new section must identify the gap in the current solutions and how their contribution will cover that gap. The introduction must remain as simple as possible, including the last paragraph about the aim of the paper. Authors can consider moving part of section 2 to the new section about state of the art.
3- The conclusions section is extremely long. The conclusion after a long discussion has to be brief and concise. Authors have to consider moving part of the conclusions to the discussion and reducing the conclusions' content. At the end of the conclusions, it is mandatory to have a paragraph detailing the future work linked to their results.
4- Given the structure and the aim of the paper, it looks like a survey or review. In these types of papers, it is common to have a section in which after analysing the current trends, authors discuss the future perspectives of the topic. Thus, I suggest including this sort of information in the discussion to endow the paper with a high-quality analysis of the current and past scenarios. Thus, I suggest dividing the conclusions into different subsections focusing on the different areas and on the future challenges and expected plans based on the lessons learned.
5- Please check the style and format of references to align with the journal's required format.
Author Response
Thank you for your comments. Our replies are in bold.
The authors focused on an interesting topic. The topic is related to the journal, and it is updated given the EU policies. Nonetheless, some changes are required to maximise the quality of the manuscript. Most of the challenges are aimed to add more information or modifying the existing information into a new format of sections to facilitate the readers with an overview of the ideas presented in this paper. Following, I include some comments:
1- I suggest changing Figure 1 with a new Figure that shows the city as a map. If possible, authors have to generate it. In this map, authors can indicate the type of land use (green areas/residential areas/industrial areas…). Having an image of the city itself does not add much value to the paper, especially if the picture does not belong to the authors. The image of the city of Amsterdam does contribute in our view to the understanding of the challenges. This is the existing infrastructure built centuries ago with narrow streets and canals and it is not going away soon. What it means is that when you would actually start to build a new city all the historic lessons can be taken into account. In other words, when Amsterdam would have been built now, the city would have a completely other design. As more or less 50% of all the cities are still to be build this century, the lesson is look before you leap.
2- The introduction should be reworked. Authors must differentiate the problem statement from the related work. In the introduction, authors have to define the problem they are facing and their objective. The content linked to the related work has to be described in a related work section. This new section must identify the gap in the current solutions and how their contribution will cover that gap. The introduction must remain as simple as possible, including the last paragraph about the aim of the paper. Authors can consider moving part of section 2 to the new section about state of the art. Done
3- The conclusions section is extremely long. The conclusion after a long discussion has to be brief and concise. Authors have to consider moving part of the conclusions to the discussion and reducing the conclusions' content. At the end of the conclusions, it is mandatory to have a paragraph detailing the future work linked to their results. We have shortened the conclusion section
4- Given the structure and the aim of the paper, it looks like a survey or review. In these types of papers, it is common to have a section in which after analysing the current trends, authors discuss the future perspectives of the topic. Thus, I suggest including this sort of information in the discussion to endow the paper with a high-quality analysis of the current and past scenarios. Thus, I suggest dividing the conclusions into different subsections focusing on the different areas and on the future challenges and expected plans based on the lessons learned. Done
5- Please check the style and format of references to align with the journal's required format. Done
Round 2
Reviewer 1 Report
First of all, I want to thank the authors for their work for the new version of their paper. The document has been restructured and some gaps that existed have been filled. In this way, the document now has a much more correct formal aspect and when it is read it sounds like a scientific document.
From a content point of view, I want to thank the authors their effort in answering all my comments. Without a doubt, the end result has been a notable improvement in the quality of the paper.
There are some aspects where the authors' answers do not coincide with my personal opinion. For example, the excessive use of references in Dutch prevents such documents from being consulted by a large part of the scientific community, which is not familiar with that language. Perhaps it would have been more practical to reduce these references or even include a brief supplementary document where the information necessary to understand these references is collected.
Also, from my point of view the number of references to the journal in which it is published is scarce. Personally, I prefer papers to be published in journals that contain other papers with references to the same topic.
In any case, I do not think that these small discrepancies invalidate the quality of the work and the results presented. Therefore, I consider that the work has the required level to be considered for publication.